# Health Kiosks in Market: A qualitative study of factors influencing the transferability of the program to a target context in Kenya

O. Jarim[1]*, K. Lydia[2]☯, M. Grace[3]☯, M. Anselimo[4]☯

1 Department of Environmental Health and Disease Control, Jomo Kenyatta University of Agriculture and Technology, Nairobi, Kenya, 2 Centre of Public Health Research, Kenya Medical Research Institute, Nairobi, Kenya, 3 Department of Community Health Nursing, Jomo Kenyatta University of Agriculture and Technology, Nairobi, Kenya, 4 Department of Food Science and Technology, Jomo Kenyatta University of Agriculture and Technology, Nairobi, Kenya

☯ These authors contributed equally to this work.
* oduoromogi@gmail.com

## Abstract

Public health kiosks can enhance access to healthcare in rural communities. To understand how to apply such an effective intervention and achieve results in another setting is key. This study explores the factors that influence the transferability of health kiosks in community markets in Kenya, with a focus on cardiovascular disease prevention and health promotion. The study was conducted in Vihiga and Nyeri counties, Kenya, between December 2023 and January 2024, among 64 participants. This qualitative exploratory study was nested in an implementation project dubbed Health Kiosks in Markets (HEKIMA). Six focus group discussions and sixteen in-depth interviews were conducted among purposively selected healthcare workers, market leaders, and policy actors. The conceptual framework, Population–intervention–environment–Transfer Model of Transferability (PIET-T), guided the study design, data analysis, and identification of key themes. It also provided recommendations to optimize the program transfer. The successful transfer of a community-based health intervention to new settings depends on four key factors: its demonstrated effectiveness, compatibility with the local infrastructure, alignment with decision-makers' priorities, and the level of institutional and stakeholder support. The findings provide valuable insights for guiding and informing the applicability of an intervention to other contexts. We demonstrate that active stakeholder participation should be encouraged.

## Introduction

With the rapid advancement of science and technology and the improvement of living standards, chronic non-communicable diseases (NCDs) such as cardiovascular diseases, cancer, diabetes, and chronic respiratory diseases have emerged as

**Data availability statement:** Data generated and analysed during this study is included in the qualitative data repository. https://data.qdr.syr.edu/dataset. xhtml?persistentId=doi:10.5064/F6EH21FD.

**Funding:** The author(s) received no specific funding for this work.

**Competing interests:** The authors have declared that no competing interests exist.

significant global public health challenges [1]. Approximately 75% of all NCD-related deaths and 85% of premature deaths (those in people aged 30–69 years) occur in low- and middle-income countries (LMICs) [2,3].

Cardiovascular disease (CVD) remains a significant cause of premature mortality [4]. Kenya faces a rapid demographic and epidemiological transition and a rising burden of CVD, resulting in a quarter of hospital admissions with 13% of autopsies identifying CVDs as the primary cause of death. Moreover, these diseases play a pivotal role in perpetuating poverty due to the financial burden of healthcare expenses, leading to high levels of out-of-pocket spending and catastrophic health expenditures [5]. The triple burden of disease, referred to as the 'overload imposed by the prevalence of NCDs and communicable diseases (CDs),' along with injuries in a community or country [6], being witnessed in LMICs like Kenya continues to overwhelm the capacity of the current health system. Recent evaluations of the capacities of several countries, including Kenya, Uganda, and Zambia, to manage NCDs have shown a considerable gap in the delivery of NCD-related services [7]. In a study to determine health service availability for NCDs in Kenya, the findings showed that only 34% of facilities surveyed were considered ready for NCD services as measured by the presence of standard precautions, basic amenities, basic equipment, and essential medicines [8]. Healthcare systems in LMICs are characterized by medication and supply shortages, a lack of monetary incentives, and underfunded healthcare structures [9]. In addition, high healthcare costs, inadequate infrastructure, and health worker shortages all decrease the capability of low- and middle-income nations to deliver primary healthcare services to their populations [10]. NCD management integration faces systematic institutional barriers, including poor policy implementation, inadequate community engagement, and limited access to and poorly integrated NCD care [9]. The above findings highlight the significant gap in low-tier health facilities, which is the primary focus of this study.

This study was nested within an ongoing community-based health intervention (CBHI) project, dubbed Health Kiosks in Market (HEKIMA), which aimed to improve access to health promotion services to prevent cardiovascular diseases in Vihiga County, Kenya [11]. HEKIMA seeks to explore whether kiosks in community markets, run by community health workers (CHWs), who are supervised by HC nurses, can improve the reach of CVD preventive care to communities. The project aligns with the task shifting guidelines by the WHO, defined as "the rational redistribution of tasks among health workforce teams" from trained and qualified health workers to other health workers with shorter training durations, to maximize the available health workforce [12].

Market users walk in for free screening, lifestyle advice, and referral to the link facility. Access to quality healthcare for populations is essential in achieving Universal Health Coverage (UHC) and the health and health-related Sustainable Development Goals (SDG [13]. This study, therefore, amplifies the process of assessing the transferability of HEKIMA from Vihiga County (primary context) to Nyeri County (target context). Transferability is the extent to which the measured impact of an applicable intervention could be achieved in another setting [14]. There is great interest in the transferability of complex public health interventions like HEKIMA, hence the need to understand the

role of context, given that an effective intervention in one setting may fail to show similar results when implemented elsewhere. HEKIMA can be considered an example of a complex public health intervention, as it is formed by interacting components that require input from multiple stakeholders, which respond flexibly to the changing needs and context [15].

This study focused on the community as a setting, target, agent, and resource. The study employed the Population-Intervention-Environment-Transfer Model of Transferability (PIET-T) in Fig 1 focusing on the perspectives of decision-makers, service providers, and other stakeholders [16].

## Materials and methods

### Study design

This was an exploratory qualitative research study. The design enabled an in-depth exploration and understanding of the participants' experiences and perspectives regarding the factors that may influence the transfer of HEKIMA from Vihiga County to Nyeri County.

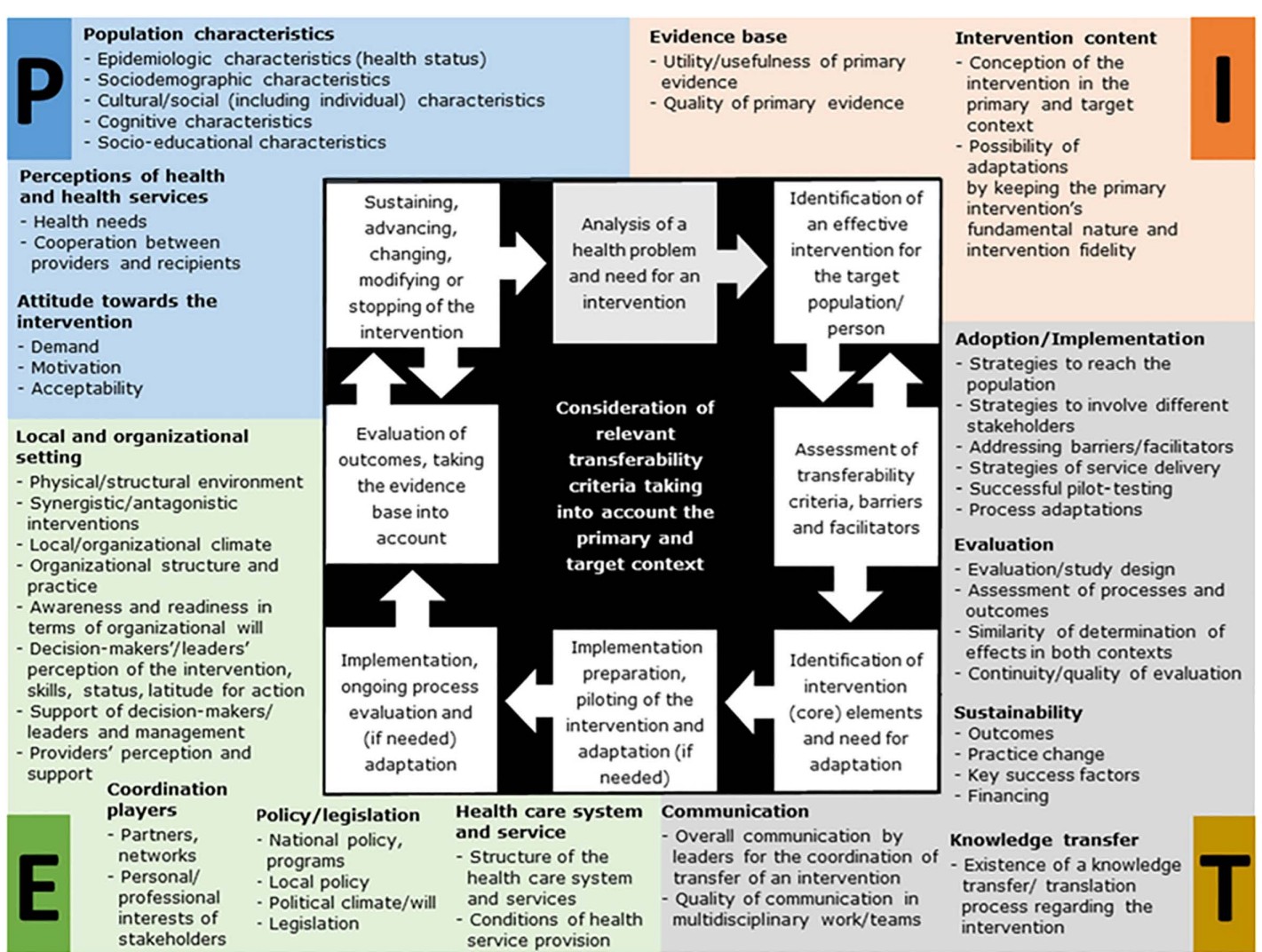

**Fig 1. A process model for the assessment of transferability (PIET-T process model).**

For this study, the health kiosks referred to herein are manned by community health promoters (CHPs) and supervised by nurses. The aim of the well-equipped and staffed Health Kiosks within markets is to encourage patients to attend regular check-ups and promote early screening [17].

## Setting

The study was conducted in Vihiga and Nyeri counties located in the western and central parts of Kenya, respectively between December 2023 and January 2024. Vihiga County has a population of approximately 600,000, compared to Nyeri's 760, [18]. The poverty level in Vihiga is approximately 38.6%, compared to 21% in Nyeri County [18]. Vihiga County has one level five health facility and three level four hospitals, with the average distance to the nearest facility being 3 kilometers (km) [19]. In Nyeri County, the distribution of health facilities in six [6] of the sub-counties is reasonably good as per WHO standards, since most of the facilities are within a radius of 1–5 km except in Kieni East and West sub-counties where health facilities are far apart ranging between 5–15 km [20]. In this study, the primary context was Vihiga County, where the intervention was conducted, while the target context was Nyeri County, where a similar intervention was to be replicated.

## Study population and sampling strategy

A total of 64 respondents aged 18 years and above were interviewed, including community health promoters (CHPs), nurses, community strategy focal persons, market champions, market chairpersons, directors of health services, and NCD coordinators. A total of six focus group discussions (FGDs) and sixteen in-depth interviews (IDIs) were done. Four FGDs were conducted in Vihiga, while two were conducted in Nyeri County. Purposive sampling of county and sub-county health management teams included the two directors of health services, two community strategy focal persons, two NCD coordinators, three market chairpersons, and two market champions. Two health facility in-charges and two nurses working at the health kiosks were purposively sampled. There were 10 community health promoters, on the other hand, identified by the community health assistants from a total of 10 community health units. The county health management team (CHMT) and sub-county health management team (SCHMT) participants were chosen based on the offices they held. The inclusion criteria for the participants were: (1) Adults aged 18 years and above, (2) people with key information on health matters in the two counties, and (3) healthcare workers and persons directly involved in setting up the health kiosks. The exclusion criteria, on the other hand, were healthcare workers from the two counties who were neither decision-makers nor had directly been involved in the implementation of the project.

## Data collection tools and procedure

Interview schedules were utilized during FGDs and IDIs. The IDIs and FGDs were conducted in December 2023 by 5 research assistants (RAs) with prior training and experience in qualitative research. The IDIs were conducted in both English and Swahili, while FGDs were conducted in Swahili. The RAs were not familiar with any of the study participants prior to conducting the interviews. Prior to study initiation, the RAs were trained for five days on the study protocol, principles of qualitative data collection and how to conduct high-quality interviews. The two qualitative guides were explicitly created for this study to ensure consistent focus on the following topics among participants: (1) the physical/structural environment of the two study sites, (2) the organizational structure of both the health system and HEKIMA, (3) the awareness and readiness of stakeholders for HEKIMA, and (4) the perspectives and support of the decision makers/leaders and service providers towards the use of HEKIMA in addressing public health issues. The interview guides were piloted in Kisumu County, but the responses were not included in the final analysis. The IDIs were conducted in the respective offices of the interviewees, while the FGDs were either held at the health kiosks or at the chosen homes, which served as the central location. Each IDI lasted for about forty five minutes, while the FGDs lasted for one hour and fifteen minutes. All interviews were recorded upon getting written consent from the participants. The recording was supplemented with

concurrent note-taking. Numbers were used to identify the participants as a way to ensure anonymity. In the case of data from the county and sub-county health management team, where it was impossible to disguise the respondent, anonymity was assured by using pseudonyms to de-identify the respondents. To ensure confidentiality, transcripts and voice recorders were transferred to a password-protected, locked laptop that was accessed only by the principal investigator.

## Data analysis

Qualitative data were transcribed and analyzed using a deductive content analysis approach. Deductive content analysis derives its labels from existing theories [21] and this is in line with our study having used a (PIET-T) model to test its applicability in the context of health kiosks in markets. NVivo was used for data organization [22]. The researcher began by creating a new project (Hekima Project) in the software. All transcribed scripts were then uploaded into the software where the author reviewed each transcript several times and independently developed 264 codes. The codes were later merged into thirteen sub-themes and fitted under the PIET-T themes as shown in Table 1

## Ethical considerations

Study approvals were obtained from the Department of Environmental Health and Disease Control and the Deanship of Graduate Studies at Jomo Kenyatta University of Science and Technology. Additionally, ethical clearance for this study was obtained from the Kenya Medical Research Institute (KEMRI) Scientific Ethics Review Unit (SERU) (KEMRI/SERU/CPHR/54-02-2023/4721). Further approval was granted by the National Commission for Science, Technology and Innovation (NACOSTI/P/23/29162). At the same time, permission was obtained from the County government of Vihiga and

Table 1. Thematic framework on the factors influencing the transferability of health kiosks in markets.

| PIET-T Domain | Sub-theme | Description |
| --- | --- | --- |
| Intervention (I) | Accessibility of kiosk location | Whether where the kiosk is located nearby, is convenient to people and easy to find. |
| | Design simplicity and usability | Whether the kiosk interface is user-friendly and accessible to diverse literacy levels |
| | Scope of services offered | Range and relevance of services (health screening, health information and referrals) in meeting community health needs. |
| | Affordability | Direct and indirect costs to users and whether the model remains financially sustainable in low-income settings. |
| | Perceived usefulness of the intervention | The community and the larger stakeholders view on the worth, value and functionality of the intervention |
| | Supply of medical products and equipment | Availability of much needed medical supplies and maintenance of already purchased equipment |
| | Decision maker's perception towards the intervention | The County's department of health view on the overall implementation of the intervention |
| | Integration with the existing health systems | How well kiosks are linked with formal healthcare facilities for referrals and follow-up. |
| | Community Engagement and Stakeholder Support | Level of buy-in from market leaders, traders, community health workers, and local government. |
| | Infrastructure and Resources at the health kiosk | Availability of electricity, space, water and human resources for smooth operation of the health kiosk |
| | Policy/legislation and regulatory support | Existence of supportive policies and health regulations to support the operation of the health kiosk |
| Transferability (T) | Sustainability mechanisms | Long-term financing options, ownership models, and potential for local capacity building |
| | Evidence of effectiveness | The community and the larger stakeholders view on the success of the project |

the Nyeri Department of Health. All participants recruited between December 4 and 15, 2023, were provided with written informed consent, which included permission to audio record the interviews. The study was explained in detail to the participants, and they were given the opportunity to ask questions and withdraw from the study at any time if they did not wish to continue with the interview.

## Results

### Characteristics of the participants

The participants included both males and females living in either Nyeri or Vihiga counties. The health workers interviewed included healthcare professionals, policy actors, community health promoters, local market leaders, and the general public. The healthcare workers included two female nurses and two female health facility in-charges. The policy actors included two male directors of health and two male strategy focal persons in charge of community work in the two counties. In addition, we had one male and one female market champions and two male NCD coordinators. At least eight (80%) of the CHVs were female with only two (20%) being male.

The highest education level of the community health promoters was high school, while for the health workers and policy actors, it was between a diploma and a master's degree.

**Theme 1: Intervention (I).**  Accessibility of kiosk location: HEKIMA's strategic location allowed traders who spent much time in the market to go for screening services. In addition, those who came to buy or sell their products also had a chance to visit the health kiosks for screening and lifestyle advice. The health kiosks also served both locals and non-locals who were not able to visit the health facilities.

A participant from the target context mentioned that it would be important to have HEKIMA implemented in their county, given the long distance and poor terrain the locals have to endure for them to access health services. Specifically, they mentioned the mountain areas and other parts of the county where health facilities are too far to reach.

Participants from the target context also highlighted the amount of time, cost of transport, and road infrastructure to the local health facilities whenever they sought health services. They therefore suggested that building health kiosks in markets that they visit regularly would improve the knowledge of the locals on NCDs. The Nyeri participants suggested that the kiosks can also be established in shopping areas and the chief's (local administration) offices due to the constant flow of people.

*"It is strategically placed at the market because the market is a busy place, and most people do not have time to go to the facility; market users, sellers, and buyers spend most of their time at the market; hence, they do not have time to go for blood pressure or weight check-ups, it is very accessible by all people." (Community Health Promoter 1- Vihiga)*

*"Having HEKIMA will be very helpful because of the geographical areas where some facilities are very far, for instance in our border with the neighboring county, there is only one dispensary on the far end of the border. Another area that really needs such an intervention is around Mount Kenya forest where health services are not accessible to the locals." (County Health Management member 2- Nyeri)*

Design simplicity and usability: Two health kiosks were established within market settings, both adapted from shipping containers and partitioned into two functional sections to resemble a small clinic. The design emphasized simplicity and accessibility, providing a compact yet functional space that could host essential health services. In Vihiga market, the kiosk was strategically positioned at the market's entry point, ensuring visibility and easy access for individuals arriving at or leaving the market. Conversely, the kiosk in Mudete was located centrally, directly facing the area where traders displayed their goods and wares. This placement enhanced its integration into the daily routines of market users, increasing the likelihood of spontaneous service uptake. Both kiosks were clearly marked and branded, with prominent graphics outlining the services offered. The straightforward design and strong visual identity contributed to their recognition as

health service points, while maintaining a level of simplicity that aligned with the resource-constrained settings in which they were implemented.

Scope of services offered: The health market kiosks provided a range of preventive and promotive services aimed at addressing cardiovascular risk factors within community settings. Routine screening was conducted for obesity and hypertension conditions known to increase susceptibility to cardiovascular disease. These screenings allowed for early identification of individuals at risk, with immediate feedback provided on-site. In addition to screening, the kiosks served as health education points, where clients received structured health promotion talks. The discussions emphasized lifestyle-related risk factors, including the harms of smoking, excessive alcohol consumption, and high dietary salt intake. Through these interactive sessions, the kiosks functioned not only as clinical screening units but also as platforms for community sensitization on healthy living. Individuals requiring further medical evaluation or treatment were promptly referred to nearby primary care facilities attached to the kiosk, thereby ensuring linkage to the formal health system. This integrated approach highlighted the kiosks' role in bridging community health promotion with clinical care pathways.

A CHW had this to say regarding the services they offer at the kiosk.

*"Many have been coming to test their blood pressure, hypertension or diabetes, though we don't screen the blood sugar, but we look at the possibility from the signs that show up. Many are suggesting that we should be screening the blood sugar and that medicine should be availed here since they would like to see this project expanding"*

Community health Worker

Affordability: The health kiosks offered screening for obesity and hypertension as well as health promotion talks addressing smoking, excessive alcohol intake, and high salt consumption, all provided at no cost to clients. The free provision of these services emerged as a major factor in attracting individuals to the kiosks, with many respondents reporting that cost had been a consistent barrier to accessing healthcare in conventional facilities. Participants noted that financial constraints often delayed their ability to seek medical attention, contributing to advanced disease presentation and complications, particularly in relation to cardiovascular conditions. By eliminating user fees, the kiosks not only increased uptake of preventive and promotive services but also positioned themselves as an accessible entry point for individuals who might otherwise forgo care.

During an FGD, a participant indicated that apart from the health kiosk being located in an accessible position, many of the residence shy away from going to other conventional facilities for fear of being charged some money.

*"This health kiosk is located in a very good place and many of us feared to go to thehospital to be tested for pressure because of its whereas the testing here is free andadvantageous to those who visit the marketplace and get to know their status."* A maleresident during an FGD

A participant from Nyeri highlighted cost as a major barrier to healthcare access, noting that older community members face compounded challenges due to both limited mobility and inability to afford services.

"*I would like to ask if we can have a facility nearby, let us be build it here. Our parents are old and they have no money, and they cannot travel all the way to the health facility. Imagine, even just testing their pressure, they have to travel all the way to Narumoru level 4.*" A female resident during an FGD

**Perceived usefulness of the intervention:** The health providers from Vihiga described how useful HEKIMA was in terms of checking their health status and linking residents to the nearest health facility. At the same time, the market leadership believed that the health kiosks helped reduce cases of diabetes and high blood pressure.

The same sentiments were shared by a health worker from Nyeri who indicated that the presence of the kiosks will facilitate screening of cardiovascular diseases, high blood pressure, and diabetes, in addition to improving the knowledge of the locals on ways to prevent NCDs. Through this, the service provider indicated that suspected cases would be referred to the health facility for further tests and management, hence reducing cases of NCDs.

Participants both in Vihiga and Nyeri noted that the health kiosk improved the knowledge and perception of the locals towards NCDs, including high blood pressure and obesity.

In Nyeri, the participants also mentioned that HEKIMA will help in the screening of high blood pressure (HBP) and diabetes among the locals who may have a phobia of hospitals.

*The majority of the patients just come as individuals to check on their blood pressure and wellness fitness. Some may have issues like persistent headaches, so they come to seek services. (Healthcare worker 3- Vihiga)*

*"The presence of Kiosks will help in screening of cardiovascular diseases, high blood pressure, and diabetes, and advise locals on prevention of NCDs."(Healthcare worker Nyeri)*

HEKIMA, according to the policymakers, identified numerous cases of HBP and diabetes based on the numbers referred. The county health team opined that the output of HEKIMA in terms of the number of cases has, to some extent, informed the next course of action, including planning for other NCD interventions.

The CHPs and the market leadership in Vihiga expressed satisfaction with HEKIMA, citing the knowledge they continue to receive on the prevention and management of NCDs. According to the participants, HEKIMA has tremendously improved their knowledge and confidence when it comes to talking to the locals on matters of cardiovascular disease.

**Theme 2: Environment (E).** Supply of medical products and equipment: Interviews with CHWs revealed limitations in the availability of essential equipment within the kiosks. In particular, blood glucose testing kits were lacking, constraining the ability to conduct comprehensive screening for diabetes. Additionally, some of the blood pressure machines supplied to the kiosks were nonfunctional due to the absence of replacement batteries, and further restricting service delivery. These gaps highlight the dependence of the health kiosk on basic medical tools to fulfill their preventive role. From the community perspective, focus group discussion participants in Mudete and Chavakali emphasized the need for the kiosks to stock medicines. While acknowledging that the kiosks were not mandated to dispense drugs, residents expressed frustration that the absence of treatment options limited the perceived value of the services offered. Together, these findings underscore the tension between the intended preventive function of the kiosks and community expectations for a more comprehensive package of care.

*"Medicines should be availed, that's what is important. Because we are going far to lookfor them."* A male resident during an FGD

Decision maker's perception towards the intervention: Participants reported positive views towards the intervention. The most valued intervention attributes were relative advantage over existing services at the local health facilities and closeness to the people, with a participant referring to it as "yielding good fruits."

According to a county health management member at the primary context, the number of locals attending the local facilities for screening and other health promotion services reduced since they could receive screening at the health kiosks. They also suggested that HEKIMA be built across the county given its benefits.

In the target context, one of the policy-actors stated that the health kiosks would be helpful, but the intervention should be structured in a way that it compliments the work that is currently being done by the CHPs who visit at least 100 households. In addition, the MoH official in Nyeri opined that it would be important to hire more CHPs so that both the households and the kiosks are attended to by these health workers.

One of the participants in Nyeri also noted that HEKIMA would be a challenge to sustain hence suggested the need to charge service fee should there be no funding from partners. According to the policy actor, the kiosks should be managed at the health unit level where it would use the existing health structure.

However, the decision-makers both in Nyeri and Vihiga pointed out that acute inadequate human resources for health, lack of medical products and equipment, and poor health system financing are some of the factors that would hinder optimum service delivery at the health kiosks in markets.

In order to improve the health system funding at the counties, a participant opined that more resources need to be shared with the county governments given that health functions were devolved. Through this, the policy maker felt that the regional governments will be able to improve on the human resource for health and avail medicine, supplies, and technologies which remain a challenge.

*"I want to say this: the national government keeps a lot of money at the national level; they need to devolve those funds so that these funds can be able to buy drugs, employ human resources for health, which is the biggest problem in our counties." (Policy maker 1-Vihiga)*

Integration with the existing health systems: Perspectives from county-level health decision makers, kiosk health workers, and community health promoters consistently highlighted the importance of linking health kiosks with the wider health system. Respondents noted that integration would strengthen cooperation between community-based services and conventional health facilities, ensuring that preventive activities conducted at the kiosks are effectively connected to diagnostic and treatment pathways in the formal system. Such alignment was seen as critical for achieving continuity of care, minimizing service duplication, and facilitating timely referrals from the community level to higher tiers of the health system. In addition, the facility will benefit from commodities and will also have a chance to get extra staff. Stakeholders emphasized that without structured integration, the kiosks risk operating in isolation, thereby limiting their potential contribution to broader health system goals.

Community engagement and stakeholder support: Our findings show that the project had support of the management from the onset during the ideation where key individuals attended all the meetings called upon by the project team. In addition, the local administration, together with the local political leadership, went further to make people aware through community gatherings of the services that were being offered at the health kiosks.

The county government of Vihiga brought in nurses who supervised the community health promoters during the implementation period. According to a policy maker, the County government, in partnership with other stakeholders, included a chapter on NCDs on key county documents, including policies and work plans. The participant hinted at the need to develop a policy that only focuses on NCDs going forward.

*"From the start, the county government has been coming in and also brought the nurses who worked together with our community health promoters." (Market leader 2)*

*"We came up with nutritional policy, and we added a whole chapter talking about NCDs. Given that NCDs are even bigger, we may think just having a policy for NCDs alone." (County Health Management Team member 1)*

A participant stressed the need for leadership support of HEKIMA. The stakeholder viewed political support to such an intervention as pivotal due to their influence and power in the community.

*"The day-to-day happenings or any development relies on the political class. So, their full support to an intervention like HEKIMA can go a long way in its service improvement and community acceptance as a whole." (Market champion 1)*

In Nyeri County, a policy actor confirmed that the current leadership in the county would be ready to embrace an intervention serving its own people. According to the participant, with the current high burden of NCDs in Nyeri, the County government was ready to welcome partners ready to assist in combating the disease. The policy actor further noted the need to involve the political class and leadership much earlier for ownership and buy in.

Infrastructure and resources: Respondents from both intervention and target sites identified several systemic challenges affecting access to conventional health facilities. Distance to facilities and poor road infrastructure were frequently cited as barriers that limited timely access to care. Even when facilities were reached, participants noted overcrowding, long waiting times, and a shortage of healthcare workers, which collectively undermined service delivery. In addition, stock-outs of essential medicines were reported as a recurrent concern. Against this backdrop, the presence of health kiosks was perceived as a valuable alternative, reducing the need for long travel and providing more immediate access to services. At the kiosk level, however, gaps in basic infrastructure were also observed. Community health workers and nurses emphasized the need for functional amenities such as toilets and improved waiting spaces to better accommodate clients. These findings highlight both the broader health system challenges and the specific infrastructural needs required to optimize service provision at the kiosks.

Policy/legislation and regulatory support: Interviews with policymakers underscored the critical role of policy frameworks in sustaining health kiosk initiatives. Respondents emphasized that without formal policy backing, the continuity of the project would remain vulnerable to shifts in political leadership, jeopardizing long-term sustainability. The presence of a policy was seen as essential in ensuring that kiosks are institutionalized within the health system and remain operational beyond changes in government. In addition, policymakers highlighted ongoing efforts to integrate non-communicable disease (NCD) priorities into broader strategic documents. For instance, in Vihiga, the county nutritional policy incorporates a chapter on NCDs and mental health, while other strategic documents such as reproductive, maternal, and child health frameworks also include provisions addressing NCDs. Nonetheless, there was consensus that the growing burden of NCDs may require a stand-alone policy to provide stronger direction and dedicated support. These insights illustrate both the recognition of policy gaps and the emerging opportunities for embedding kiosks within existing and future health strategies.

"*While NCDs are increasingly incorporated into existing health policies, their rising impact suggests the need for a dedicated policy framework.*" Policy maker 2-Vihiga

**Theme 3: Transferability (T).** Sustainability mechanisms: The primary function of the County Departments of Health where HEKIMA is established is promoting primary health care, among others. Our findings show that there is empowerment of CHPs, the market leadership, and the nurses. In collaboration with the County Government of Vihiga, the project team trained the CHPs on symptoms, modifiable risk factors (smoking, diet, PA, alcohol, BMI, waist circumference, blood pressure, and glucose measures), stress, and motivational engagement strategies. The CHPs were also trained to promote adherence to medication and adoption of healthy lifestyle habits via monthly group-based motivation sessions and short message service (SMS) reminders. The CHPs additionally received training on communication, data management, and technology, where they learnt about the use of tablets in data collection and transfer of health kiosk's data by CHPs to nurses at the health center (HC) using a secure data transfer channel.

The project also trained the market champions on the study processes, CVD prevention, and activism to promote engagement with the kiosk. Market champions were selected in close consultation with the relevant County Department.

"*Personally, I did not know how someone with or without these conditions were to be helped, but now I am in a position to explain to someone about any matter on weight, sugar, high blood pressure, and even diet.*"(Community health promoter 2)

*As a community health promoter, my role is to receive the clients, give health education, sample, and screen. I also refer cases to the health facilities. I have been able to learn so much as a result of the project. (Community health promoter 3)*

A service provider in Nyeri suggested that the local administration, including the village elders, chiefs, and community members, should be empowered during the project's inception. According to the participant, empowering the local stakeholders would help communicate the intervention to the community. Communication on the intervention should be clear and owned by the local stakeholders.

Evidence of effectiveness: Respondents consistently described the health kiosks as effective platforms for promoting community awareness and early detection of non-communicable diseases (NCDs). The kiosks were frequently likened to "wellness clinics," where community members could routinely check their blood pressure and blood glucose levels, and receive guidance on appropriate next steps if abnormal results were detected. Market days were reported as particularly high-traffic periods, with many individuals intentionally visiting the kiosks to monitor their health status. Beyond individual service use, the kiosks were seen as strengthening the continuum of care. Community health workers (CHWs) were noted to play a critical role in following up with clients, while healthcare providers at the kiosks facilitated referrals to linked health facilities for further management. Respondents highlighted that this model not only improved access to health education but also supported timely intervention for those at risk. Overall, the kiosks were widely perceived as having made a significant contribution to community-based NCD prevention and control in Vihiga County, with participants affirming their effectiveness through both observed utilization and health outcomes.

*"Those health kiosks are equivalent to wellness clinics whereby people visit, they are able to know their pressure, their sugars, and what to do in case one is diagnosed with hypertension."* Health worker -Vihiga

*"I can assure you those clinics have been of much help, especially during market days, as people come in to check their pressure and sugars. The community has really utilized the services brought to them through the kiosks."* Policy maker 2-Vihiga

## Discussion

Perception and support from both decision-makers and service providers, the structure of the healthcare system and services, and the empowerment of all stakeholders are key factors to consider for the transferability and sustainability of an intervention in a target context. This is the first known study to utilize the process model for the assessment of health interventions derived from the analysis that includes descriptive themes and criteria of the population (P), intervention (I), environment (E), and transferability (T) of a health kiosk in markets for prevention and control of cardiovascular diseases.

The decision makers' and service providers' perceptions of the intervention were highlighted as important. For instance, perceived usefulness was linked to utilization of the intervention by the locals, which is similar to [23,24]. The decision makers from the target context shared the same opinion, noting that they would be more than willing to support interventions that directly serve the community. Perception of the decision makers depends on the benefit of the intervention to the community, which includes knowledge increase, health services offered by the intervention, and interventions aligned to local policies. For instance, the HEKIMA intervention was in line with the community health strategy 2020–2025 that intended to build the capacity of individuals and households to know and progressively realize their rights to equitable, good quality health care [25] and demand services as provided for in the Kenyan constitution 2010 [26]. This implies that if decision makers consider a valuable intervention to the community, they are likely to accept and interact with it hence sustainability. In transferring an intervention to another context, ensuring that the decision makers, the community, and all stakeholders perceive the intervention as valuable and essential is crucial.

This study further adds to the body of knowledge of the critical role of decision-makers and service providers in supporting community interventions. The transferability of HEKIMA to a target context and its success depends on the decision-makers' support that includes developing public health policies, providing resources, communicating the intervention to the community, and creating an enabling environment for the implementation of community interventions. The service providers, on the other hand, are responsible for the implementation of the intervention [27]. Complex public health interventions with active participation from all actors are more effective and sustainable than those imposed externally [28].

Engaging the already existing health system structure is a major facilitator to the transferability of complex public health interventions as it improves buy-in from the policy actors and policymakers. Linking HEKIMA with the local health facility was helpful due to the referral mechanisms.

Referral processes are known to be central to coordinating care. Care coordination is described as the intentional organization of patient care across multiple participants, accompanied by the sharing of relevant information, to improve the safety and effectiveness of services [29].

Effective referral systems from the community to the health care facility are essential to save lives and ensure both the continuity and quality of care; they can also influence community health promoters' performance [30]. Community facility linkage interventions are known to improve community health in low-income settings like Vihiga and Nyeri by strengthening a relationship between community members and the local healthcare providers [31]. There is, however, a need to improve the health infrastructure, drugs and supplies, and human resources for health in these institutions. A well-structured and functioning referral system may also serve as a motivating factor for community health promoters since it ensures a continuum of care for the clients, thus strengthening the relationship between the community and the service providers at the health kiosk.

The transferability of a health intervention should incorporate local stakeholders and their leadership to achieve a better understanding of the structural challenges, including geographical access to local facilities and an appropriate site for developing the intervention. Poor infrastructure, combined with a lack of transportation, is a barrier to receiving care in the local community [32]. Timely access to healthcare is essential, as it enables providers to prevent illness, control acute episodes, and manage chronic conditions, thereby reducing the risk of exacerbation or complications associated with these health conditions. People cannot access care if it does not exist in their geographic area; hence, access to health service sites, including health kiosks in markets, requires integration into the healthcare system, getting access to sites of care where patients can receive needed services, and finding providers who meet the needs of patients and with whom patients can develop a relationship based on mutual communication and trust [33]. This implies that complex public health interventions should be targeted more closely at the relevant groups. The interventions should also consider a locality where the community meets regularly, such as a trading center, a chief's office, or a local church. Careful consideration of local context is crucial to avoid exacerbating inequities.

## Conclusion

A model emerged from our data depicting multilevel contextual factors believed to shape the transferability of an intervention like HEKIMA. HEKIMA is also an innovation that can be implemented in combating other public health concerns.

A targeted sensitization session is needed for all stakeholders, including community leaders, health workers, and local officials, to ensure they clearly understand the value and relevance of the intervention. Empowering frontline implementers and beneficiaries, including community health promoters, nurses, and the general public, by involving them in the planning, adaptation, and implementation process is key to building ownership and local capacity. Leveraging and adapting the existing infrastructure and health services to minimize additional resource needs and increase sustainability should be encouraged. There is also a need to establish structured, multi-sectoral partnerships with clearly defined roles among stakeholders and partners implementing complex public health interventions, such as HEKIMA, to coordinate implementation, resource sharing, and problem-solving.

The national and county government should also assess and address local health system challenges upfront, including workforce capacity, service delivery gaps, and logistical barriers, to ensure complex public health interventions are realistically integrated and scaled.

Further research is needed to explore effective methods for engaging stakeholders, particularly at the community and frontline levels, specifically in the co-creation and contextual adaptation of complex public health interventions, such as HEKIMA. Such studies should evaluate how different engagement strategies influence CBHI ownership, implementation fidelity, and long-term sustainability, particularly in resource-constrained or decentralized health systems.

## Strengths and limitations

One of the strengths of our study is its use of data from multiple sources. The data analysis combined with the PIET transferability model elicited information considering a broad range of factors key to the transferability of CBHIs.

One significant limitation of our study is that participants largely self-reported on their perception of the intervention. Multiple data sources were used to validate some of this information; however, we cannot rule out the possibility of potential biases.

## Acknowledgments

The authors thank the County governments of Vihiga and Nyeri for approving the conduct of the study at the study sites. We also extend our appreciation to the respondents for their time and valuable contributions. We also appreciate all the research assistants who assisted in data collection.

## Author contributions

**Conceptualization:** Jarim Omogi.

**Formal analysis:** Jarim Omogi.

**Investigation:** Jarim Omogi.

**Methodology:** Jarim Omogi.

**Supervision:** Lydia K, Grace M, Anselimo M.

**Validation:** Lydia K, Grace M.

**Writing – original draft:** Jarim Omogi.

**Writing – review & editing:** Jarim Omogi, Lydia K, Grace M, Anselimo M.

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
