## [Decision Letter · Decision Letter 0]

26 Jun 2025

Dear Dr. Omogi,

Thank you for submitting your manuscript to PLOS ONE. After careful consideration, we feel that it has merit but does not fully meet PLOS ONE’s publication criteria as it currently stands. Therefore, we invite you to submit a revised version of the manuscript that addresses the points raised during the review process.

We look forward to receiving your revised manuscript.

Kind regards,

Tatchalerm Sudhipongpracha

Academic Editor

PLOS ONE

Journal Requirements:

Reviewers' comments:

Reviewer's Responses to Questions

**Comments to the Author**

1. Is the manuscript technically sound, and do the data support the conclusions?

Reviewer #1: Partly

Reviewer #2: Yes

Reviewer #3: Yes

2. Has the statistical analysis been performed appropriately and rigorously?

Reviewer #1: No

Reviewer #2: N/A

Reviewer #3: I Don't Know

3. Have the authors made all data underlying the findings in their manuscript fully available?

Reviewer #1: Yes

Reviewer #2: Yes

Reviewer #3: Yes

4. Is the manuscript presented in an intelligible fashion and written in standard English?

Reviewer #1: No

Reviewer #2: Yes

Reviewer #3: Yes

Reviewer #1: 1. The abstract lacks specificity in key findings. What exact factors (beyond generalities like "perceived usefulness") influenced transferability? Rewrite the results section of the abstract.

2. The introduction is overloaded with general background on NCDs; it could be summarized and added to with more Kenya-specific context and the rationale for choosing HEKIMA.

3. The flow between global issues and local intervention is not coherent.

4. In line 111, "Error! Reference source not found" suggests a missing figure or citation, indicating poor proofreading or reference management.

5. Lines 133–153 repeat participants in multiple ways and have too much redundancy in the sampling strategy.

6. Line 181 states that "inductive content analysis" (ICA) was used, yet the coding process seems highly structured and deductive, given the PIET-T framework guidance. Clarify whether it was inductive or deductive.

7. Lack of sufficient discussion on why certain elements supported or hindered transferability. For example, why did decision-makers support it in one county but have reservations in another?

8. There are numerous grammatical errors, run-on sentences, and awkward phrasings throughout the manuscript. There are inconsistent references and punctuation (e.g., missing full stops and misplaced commas). The manuscript should be proofread thoroughly.

Reviewer #2: Recommendations for Authors

Thank you for the opportunity to review the manuscript titled “Health Kiosks in market: A qualitative study of factors influencing the transferability of the program to a target context in Kenya”. While the topic is relevant and timely, I have several concerns regarding the manuscript’s current form. Below, I outline specific areas where the manuscript could be strengthened to improve its clarity, coherence, and overall contribution.

Abstract

- Please include the data collection period and the total number of participants.

- Kindly add 3-5 relevant keywords to support indexing and improve discoverability.

Introduction

- When referring to the disease burden or epidemiological situation, please specify the year to provide a clear temporal context. Also, consider updating the data if more recent statistics are available.

- The phrase “in Error! Reference source not found” appears to be a referencing error. Please review and correct the citation accordingly.

Research Methods and Design

Setting

- When stating the population size, please indicate the year the data refers to in order to ensure clarity and relevance.

- Including a map of the study area would enhance the reader’s understanding of the geographical context.

Study Population and Sampling Strategy

- Please specify the exclusion criteria.

Results

- Please provide the characteristics of participants for FGDs and IDIs to offer context for the findings.

Discussion

- Please include recommendations for further research to guide future work in this area.

- Kindly expand the discussion to provide deeper analysis or reflection on key findings and their implications for program transferability.

Acknowledgment

- Please include a note of appreciation to the study participants for their time and valuable contributions.

Figure

- Please ensure that Figure 1 is properly cited within the main text to maintain consistency and guide the reader.

- Figure 1 appears unclear and lacks sharpness. Please consider enhancing the resolution or quality to improve readability.

Reviewer #3: I am not sure, the researchers use the methodology of content analysis will help answer the question of factors influencing the transferability of the program to a target context in Kenya because the researcher used focus group and in-depth interviews. The study of factors should use quantitative methods, which should make the researcher's explanation more reliable.

Researchers should elaborate on what Health Kiosks in Market (HEKIMA) can provide. Is it specific to NCDs, what is the scope?

**Do you want your identity to be public for this peer review?** For information about this choice, including consent withdrawal, please see our Privacy Policy

Reviewer #1: No

Reviewer #2: **Yes: ** Uma Langkulsen

Reviewer #3: No

---

## [Author Response · Author response to Decision Letter 1]

7 Jul 2025

Recommendations for Authors Authors comment page

Additional Requirements

Please ensure that your manuscript meets PLOS ONE's style requirements, including those for file naming. The document has been redone and currently meets PLOS ONE’s style Across the whole document

We note that you have indicated that there are restrictions to data sharing for this study. For studies involving human research participant data or other sensitive data, we encourage authors to share de-identified or anonymized data. However, when data cannot be publicly shared for ethical reasons, we allow authors to make their data sets available upon request. For information on unacceptable data access restrictions, please see • The author has sent the data to Qualitative Data Repository

22

If there are ethical or legal restrictions on sharing a de-identified data set, please explain them in detail (e.g., data contain potentially identifying or sensitive patient information, data are owned by a third-party organization, etc.) and who has imposed them (e.g., a Research Ethics Committee or Institutional Review Board, etc.). Please also provide contact information for a data access committee, ethics committee, or other institutional body to which data requests may be sent.

If there are no restrictions, please upload the minimal anonymized data set necessary to replicate your study findings to a stable, public repository and provide us with the relevant URLs, DOIs, or accession numbers.

Please review your reference list to ensure that it is complete and correct. If you have cited papers that have been retracted, please include the rationale for doing so in the manuscript text or remove these references and replace them with relevant current references. Any changes to the reference list should be mentioned in the rebuttal letter that accompanies your revised manuscript. If you need to cite a retracted article, indicate the article’s retracted status in the References list and also include a citation and full reference for the retraction notice. I have added two references, among them Nos 5 and 21 having updated the literature and the type of analysis used. 20 & 21

Reviewer #1

The abstract lacks specificity in key findings. What exact factors (beyond generalities like "perceived usefulness") influenced transferability? Rewrite the results section of the abstract.

The results have been rewritten and the findings made specific 2

The introduction is overloaded with general background on NCDs; it could be summarized and added to with more Kenya-specific context and the rationale for choosing HEKIMA. More Kenya specific has been added into the introduction section. 3

The flow between global issues and local intervention is not coherent Inconsistencies redone on the instruction section.

In line 111, "Error! Reference source not found" suggests a missing figure or citation, indicating poor proofreading or reference management. This is done

Lines 133–153 repeat participants in multiple ways and have too much redundancy in the sampling strategy The study population and sampling strategy has been rewritten and the redundancy sorted 6 & 7

Line 181 states that "inductive content analysis" (ICA) was used, yet the coding process seems highly structured and deductive, given the PIET-T framework guidance. Clarify whether it was inductive or deductive The has been revised as per the advice given 8

Lack of sufficient discussion on why certain elements supported or hindered transferability. For example, why did decision-makers support it in one county but have reservations in another This is done

There are numerous grammatical errors, run-on sentences, and awkward phrasings throughout the manuscript. There are inconsistent references and punctuation (e.g., missing full stops and misplaced commas). The manuscript should be proofread thoroughly. The manuscript has been proofread and is now okay for publication

Reviewer #2

Abstract

Please include the data collection period and the total number of participants. This has been indicated on the abstract 1

Kindly add 3-5 relevant keywords to support indexing and improve discoverability. This has been added 1

Introduction

When referring to the disease burden or epidemiological situation, please specify the year to provide a clear temporal context. Also, consider updating the data if more recent statistics are available This is done

The phrase “in Error! Reference source not found” appears to be a referencing error. Please review and correct the citation accordingly. This has been corrected

Research Methods and Design

When stating the population size, please indicate the year the data refers to in order to ensure clarity and relevance. This has been stated 6

Including a map of the study area would enhance the reader’s understanding of the geographical context. A map has been done

Please specify the exclusion criteria Exclusion criteria has been added

Results

Please provide the characteristics of participants for FGDs and IDIs to offer context for the findings. This has been provided in the biggening of the finding section 9

Discussion

Please include recommendations for further research to guide future work in this area. This has been included as advised 19

Kindly expand the discussion to provide deeper analysis or reflection on key findings and their implications for program transferability. This is done

Acknowledgment

Please include a note of appreciation to the study participants for their time and valuable contributions. A note of appreciation to the participants added 20

Figure

Please ensure that Figure 1 is properly cited within the main text to maintain consistency and guide the reader. This has been clearly cited

Figure 1 appears unclear and lacks sharpness. Please consider enhancing the resolution or quality to improve readability. I have tried to enhance the resolution and sharpness of the figure Figures

Reviewer #2

I am not sure, the researchers use the methodology of content analysis will help answer the question of factors influencing the transferability of the program to a target context in Kenya because the researcher used focus group and in-depth interviews. The study of factors should use quantitative methods, which should make the researcher's explanation more reliable. This has been done as advised. 6

---

## [Decision Letter · Decision Letter 1]

27 Aug 2025

Dear Dr. Omogi,

We look forward to receiving your revised manuscript.

Kind regards,

Tatchalerm Sudhipongpracha

Academic Editor

PLOS ONE

Journal Requirements:

Reviewers' comments:

Reviewer's Responses to Questions

**Comments to the Author**

Reviewer #1: All comments have been addressed

Reviewer #2: All comments have been addressed

2. Is the manuscript technically sound, and do the data support the conclusions?

Reviewer #1: Yes

Reviewer #2: Yes

3. Has the statistical analysis been performed appropriately and rigorously?

Reviewer #1: Yes

Reviewer #2: N/A

4. Have the authors made all data underlying the findings in their manuscript fully available?

Reviewer #1: Yes

Reviewer #2: Yes

5. Is the manuscript presented in an intelligible fashion and written in standard English?

Reviewer #1: Yes

Reviewer #2: Yes

Reviewer #1: The manuscript is much improved and suitable for publication after addressing the issues stated.

While the authors mention developing 264 codes and merging them into 13 sub-themes, a brief mention of intercoder reliability will be beneficial. Even though optional, a table summarizing the final themes/sub-themes under the PIET-T framework would aid readers' understanding and strengthen the presentation of results.

Reviewer #2: Thank you for revising the section on participant characteristics. While the current description offers helpful qualitative details, it would be more informative to also include basic quantitative information, such as the number of participants in each group (e.g., how many FGD and IDI participants, by role or gender) to better contextualize the findings.

**Do you want your identity to be public for this peer review?** For information about this choice, including consent withdrawal, please see our Privacy Policy

Reviewer #1: No

Reviewer #2: **Yes: ** Uma Langkulsen

---

## [Author Response · Author response to Decision Letter 2]

1 Sep 2025

Reviewer #1: The manuscript is much improved and suitable for publication after addressing the issues stated.

While the authors mention developing 264 codes and merging them into 13 sub-themes, a brief mention of intercoder reliability will be beneficial.

While intercoder reliability is key in measuring consistency and replicability among others, it can only be applied when qualitative has been analyzed by more than one coder. This being a PhD, the researcher did not engage the services of other analysts hence it may not be possible to develop the reliability test.

Even though optional, a table summarizing the final themes/sub-themes under the PIET-T framework would aid readers' understanding and strengthen the presentation of results.

A table summarizing the thirteen themes have been added in page 8.

Reviewer #2: Thank you for revising the section on participant characteristics. While the current description offers helpful qualitative details, it would be more informative to also include basic quantitative information, such as the number of participants in each group (e.g., how many FGD and IDI participants, by role or gender) to better contextualize the findings.

This information has been captured in the study population section. I have also added more information in the results section that is found in page 9 and 10.

---

## [Decision Letter · Decision Letter 2]

11 Sep 2025

Health Kiosks in Market: A qualitative study of factors influencing the transferability of the program to a target context in Kenya.

PONE-D-25-11656R2

Dear Dr. Omogi,

We’re pleased to inform you that your manuscript has been judged scientifically suitable for publication and will be formally accepted for publication once it meets all outstanding technical requirements.

Kind regards,

Tatchalerm Sudhipongpracha

Academic Editor

PLOS ONE

Additional Editor Comments (optional):

Reviewer #2:

Reviewers' comments:

Reviewer's Responses to Questions

**Comments to the Author**

Reviewer #2: All comments have been addressed

2. Is the manuscript technically sound, and do the data support the conclusions?

Reviewer #2: Yes

3. Has the statistical analysis been performed appropriately and rigorously?

Reviewer #2: N/A

4. Have the authors made all data underlying the findings in their manuscript fully available?

Reviewer #2: Yes

5. Is the manuscript presented in an intelligible fashion and written in standard English?

Reviewer #2: Yes

Reviewer #2: Thank you for the opportunity to review the manuscript titled “Health Kiosks in Market: A qualitative study of factors influencing the transferability of the program to a target context in Kenya”. The authors have addressed the reviewer’s comment. I wish them the best of luck with the manuscript.

**Do you want your identity to be public for this peer review?** For information about this choice, including consent withdrawal, please see our Privacy Policy

Reviewer #2: **Yes: ** Uma Langkulsen

---

## [Editor Report · Acceptance letter]

PONE-D-25-11656R2

PLOS ONE

Dear Dr. Omogi,

I'm pleased to inform you that your manuscript has been deemed suitable for publication in PLOS ONE. Congratulations! Your manuscript is now being handed over to our production team.

Kind regards,

on behalf of

Dr. Tatchalerm Sudhipongpracha

Academic Editor

PLOS ONE